# Internalization and Transport of PEGylated Lipid-Based Mixed Micelles across Caco-2 Cells Mediated by Scavenger Receptor B1

**DOI:** 10.3390/pharmaceutics13122022

**Published:** 2021-11-26

**Authors:** Xiangjie Su, Mercedes Ramírez-Escudero, Feilong Sun, Joep B. van den Dikkenberg, Mies J. van Steenbergen, Roland J. Pieters, Bert J. C. Janssen, Peter M. van Hasselt, Wim E. Hennink, Cornelus F. van Nostrum

**Affiliations:** 1Department of Pharmaceutics, Utrecht Institute for Pharmaceutical Sciences, Utrecht University, 3584 CG Utrecht, The Netherlands; x.su1@uu.nl (X.S.); dragonfly88214@hotmail.com (F.S.); j.b.vandendikkenberg@uu.nl (J.B.v.d.D.); m.j.vansteenbergen@uu.nl (M.J.v.S.); w.e.hennink@uu.nl (W.E.H.); 2Structural Biochemistry, Bijvoet Center for Biomolecular Research, Faculty of Science, Utrecht University, 3584 CG Utrecht, The Netherlands; mramirezescudero@gmail.com (M.R.-E.); b.j.c.janssen@uu.nl (B.J.C.J.); 3Department of Chemical Biology & Drug Discovery, Utrecht Institute for Pharmaceutical Sciences, Utrecht University, 3584 CG Utrecht, The Netherlands; R.J.Pieters@uu.nl; 4Department of Pediatrics, Wilhelmina Children’s Hospital, University Medical Center Utrecht, 3584 EA Utrecht, The Netherlands; P.vanhasselt@umcutrecht.nl

**Keywords:** vitamin K, mixed micelles, PEGylation, scavenger receptor B1, epithelial transport

## Abstract

The aim of this study was to get insight into the internalization and transport of PEGylat-ed mixed micelles loaded by vitamin K, as mediated by Scavenger Receptor B1 (SR-B1) that is abundantly expressed by intestinal epithelium cells as well as by differentiated Caco-2 cells. Inhibition of SR-B1 reduced endocytosis and transport of vitamin-K-loaded 0%, 30% and 50% PEGylated mixed micelles and decreased colocalization of the micelles with SR-B1. Confocal fluorescence microscopy, fluorescence-activated cell sorting (FACS) analysis, and surface plasmon resonance (SPR) were used to study the interaction between the mixed micelles of different compositions (varying vitamin K loading and PEG content) and SR-B1. Interaction of PEGylated micelles was independent of the vitamin K content, indicating that the PEG shell prevented vitamin K exposure at the surface of the micelles and binding with the receptor and that the PEG took over the micelles’ ability to bind to the receptor. Molecular docking calculations corroborated the dual binding of both vita-min K and PEG with the binding domain of SR-B1. In conclusion, the improved colloidal stability of PEGylated mixed micelles did not compromise their cellular uptake and transport due to the affinity of PEG for SR-B1. SR-B1 is able to interact with PEGylated nanoparticles and mediates their subsequent internalization and transport.

## 1. Introduction

Polyethylene glycol (PEG) is a hydrophilic polymer, which is widely used to decorate the surface of nanoparticles, called PEGylation, to improve their colloidal stability, avoid aggregation, and improve mucus penetration [1,2,3,4]. Recently, we developed PEGylated mixed micelles that retain their stability at acidic conditions and exhibited better mucus penetration than corresponding non-PEGylated formulations (i.e., Konakion MM^®^) [5]. Konakion^®^ MM is a commercial formulation of vitamin K based on mixed micelles composed of egg phosphatidylcholine (EPC) and glycocholic acid, which is indicated for prophylaxis and treatment of vitamin K deficiency bleeding (VKDB) in neonates as well as infants [6,7]. This fat-soluble vitamin is present in food (e.g., plants) and is taken up after solubilization in the intestine into mixed micelles of bile acids and phospholipids in healthy individuals. However, in neonates suffering from cholestasis, the colloidal instability and aggregate formation of this formulation in the stomach due to the low pH prevent the absorption of vitamin K [6,8]. In our previous study, DSPE-PEG 2000 was introduced as a micellar component to improve both the colloidal stability at low pH and mucus diffusion [9]. The PEGylated micelles still showed, although reduced, uptake and transport of vitamin K by Caco-2 epithelial cells as compared to their non-PEGylated counterparts [9]. Importantly, a recent preclinical study demonstrated that the developed PEGylated mixed micelles showed adequate and reliable intestinal absorption of vitamin K under cholestatic conditions in rats [10]. 

It was shown in our previous study that the Scavenger Receptor B1 (SR-B1) is most likely involved in endocytosis of vitamin-K-loaded mixed micelles with and without PEGylation [9]. The interaction between vitamin K and SR-B1 has been demonstrated previously [11]; however, the mechanism of interaction between the mixed micelles and SR-B1 was not investigated, which motivated us to investigate the relationship between the degree of PEGylation of the mixed micelles and the involvement of SR-B1 in the uptake and transport. 

SR-B1 is a well characterized transmembrane 509 amino acid protein, with a large extracellular loop (~408 residues) containing multiple sites that can bind a variety of ligands, including high density lipoprotein (HDL) [12], phospholipids [13], carotenoids [14], and vitamins [15]. SR-B1 consists of a heavily glycosylated and fatty acylated protein backbone, which contains beside the large extracellular loop, two transmembrane domains, and short intracellular N-terminal and C-terminal domains [14]. Research using SR-B1 knockout mice provided evidence that SR-B1 is required for formation of microvilli and microvillar channels [16]. Various approaches have been utilized to identify the structural features important for the function of SR-B1 in the interaction and uptake of the above-mentioned ligands. Interestingly, a very recent publication from Raith et al. indicated, for the first time, the affinity of PEG with SR-B1, when they demonstrated that soft, flexible, elongated nanoparticles (so-called filomicelles) with PEG exteriors and poly-butadiene (PBD) interiors were able to interact with SR-B1 receptor, typically with a ~ 6-fold stronger interaction than PEG-PBD spheres [17]. There have been a few papers published in which the binding of lipid-based nanoparticles with SR-B1 was investigated [18,19]. At present, it is not clear how endocytosis and transport of vitamin-K-loaded in mixed micelles is mediated by SR-B1. More specifically, it is unknown how the micellar components including PEGylation affect the interaction with SR-B1 and which domain of the receptor is responsible for trapping of the micelles. 

In the current study, we investigated the role of SR-B1 in the binding, internalization, and transport of vitamin-K-loaded mixed micelles with and without PEGylation. Therefore, SR-B1 was overexpressed in Hela cells to figure out its role in binding and internalization of mixed micelles. Human colon carcinoma cell cultures (Caco-2) as intestinal epithelium model were applied to investigate the colocalization of vitamin-K-loaded mixed micelles with SR-B1. Finally, the interaction was investigated by surface plasmon resonance (SPR) and molecular docking to study the binding with and affinity for mixed micelles and SR-B1. 

## 2. Materials and Methods

### 2.1. Materials

Egg phosphatidylcholine (EPC) and 1,2-distearoyl-sn-glycero-3-phosphoethanolamine-*N*-[methoxy (polyethylene glycol)] 2000 (DSPE-PEG 2000) were purchased from Lipoid GmbH (Ludwigshafen, Germany). 1,2-Dipalmitoyl-sn-glycero-3-phosphoethanolamine-*N*-(lissamine rhodamine B sulfonyl) (rhodamine conjugated PE) was obtained from Avanti Polar Lipid, Inc. (Alabaster, AL, United States). Chloroform, methanol, and ethanol were provided by Biosolve, Valkenswaard, the Netherlands. A primary antibody against Scavenger Receptor B1 (mouse anti human CLA-1) was a product of BD bioscience, Vianen, The Netherlands. Lipofectamine 3000, Alexa 488-labeled goat anti-mouse IgG, and Alexa 555-labeled goat anti-mouse IgG were obtained from Thermal Fisher Scientific (Waltham, MA, United States). Recombinant SR-B1 (78 kDa, extracellular domain Pro 33-Tyr 443, purity > 95%) and a plasmid containing SRB1 cDNA with C-terminal green fluorescence (GFP) tag were from Sino Biological (Chesterbrook, PA, United States). Zinc powder (60–70 μm) was provided by Alfa Aesar, Tewksbury, MA, United States, *N*-(3-dimethylaminopropyl)-*N*’-ethylcarbodiimide hydrochloride (EDC), *N*-hydroxysuccinimide (NHS), 2-(*N*-morpholino) ethanesulfonic acid (MES), ethanolamine hydrochloride, Dulbecco’s modified eagle’s medium (DMEM, 4.5 g/mL glucose), Dulbecco’s phosphate buffered saline (PBS), non-essential amino acids (NEAA), fetal bovine serum (FBS), Hank’s balanced salt solution (HBSS), vitamin K, glycocholic acid hydrate, bovine serum albumin (BSA), paraformaldehyde, saponin, Trion X100, Fluoroshield™ solution containing DAPI, EDTA/trypsin solution (0.05% trypsin, 0.02% EDTA), oleic acid, block lipid transport-1(BLT-1) and fluorescein isothiocyanate (FITC)–dextran, and all other chemicals and reagents were purchased from Sigma-Aldrich (Zwijndrecht, The Netherlands). All chemicals and solvents were used as received. 

### 2.2. Preparation and Characterization of Mixed Micelles of Different Compositions

Mixed micelles were prepared as described previously [9]. Briefly, glycocholic acid hydrate (110 mg, 0.24 mmol) was dissolved in a mixture of 1.5 mL methanol and 3.5 mL chloroform, together with various amounts of vitamin K and lipids (Table 1). Next, films were obtained after evaporation of the solvents of 5 mL mixture introduced in a round bottom flask (50 mL) under reduced pressure and were subsequently hydrated in 8 mL phosphate buffer (25 mM KH_2_PO_4_ and 42 mM Na_2_HPO_4_·2H_2_O, pH 7.3). After magnetically stirring for at least 4 h at room temperature, the obtained dispersions were pressed 3 times through a nylon syringe filter (0.2 μm, Phenomenex, Torrance, CA, USA). Micelles with different PEG contents (0, 15%, 30% or 50%) but with same loading (20 mg) of vitamin K were prepared (Table 1, 20VK-nonPEG, 20VK-15%PEG, 20VK-30%PEG and 20VK-50%PEG, respectively). Corresponding micelles of the same lipid compositions but without vitamin K loading are called empty micelles and were obtained using the same procedure. Non-PEGylated micelles with lower amounts of vitamin K (5 or 10 mg) were prepared as well (Table 1, 10VK-nonPEG and 5VK-nonPEG, respectively). For preparation of rhodamine-labeled vitamin K-loaded mixed micelles, the same method was used but the lipid composition was supplemented with 260 μg rhodamine-conjugated PE. The Z-average size of different mixed micelles was measured by dynamic light scattering (DLS; Zetasizer Nano S, Malvern Instruments, Worcestershire, UK) at 25 °C. Dispersions were diluted 20 times with 10 mM HEPES buffer (pH 7.4), and the zeta-potential of the different micelles was determined by Zetasizer Nano Z (Malvern Instruments Ltd., Worcestershire, UK). To determine the concentrations of vitamin K in the dispersions of the mixed micelles, 40 μL micelles were mixed with 360 μL ethanol and vortexed for 5 min. After centrifuging 10 min at 8000 *g* at room temperature, 20 μL supernatant of the different samples was injected onto a Sunfire C18 HPLC column with 95% ethanol as mobile phase at 1 mL/min. Encapsulation efficiency (EE%) and loading % were calculated based on following equations, respectively: EE% = Encapsulated vitamin K/Total vitamin K(1)
LC% = Weight of encapsulated vitamin K/Weight of micelle(2)

### 2.3. Cell Culture 

Caco-2 (ATCC HTB-37) cells and Hela (ATCC CCL-2) cells were obtained from American Type Culture Collection (ATCC, Manassas, VA, USA) and cultured in DMEM supplemented with 10% FBS and 1% NEAA in an incubator at 37 °C with 5% CO_2_. Penicillin and streptomycin (both 100 U/mL) were added to the medium for differentiated Caco-2 cells (21 days culture) to prevent bacterial growth. Hela cells were cultured under the same condition with DMEM supplemented with 10% FBS as medium. 

### 2.4. Transport of Vitamin-K-Loaded Mixed Micelles with Different PEG Densities through Differentiated Caco-2 Monolayers

Caco-2 cells were seeded in transwell plates (polyester membrane transwell insert with 0.4 micron pore size in 24-well, Corning, Tewksbury, MA, USA) at density of 10^5^ cells per insert and cultured for 3 weeks at 37 °C with 5% CO_2_. The medium for differentiated Caco-2 cells is described in Section 2.3, which was refreshed every 2 days. The transepithelial electrical resistance (TEER) values were recorded by Epithelial Volt Ohm Meter (EVOM2, World 284 Precision Instruments, Sarasota, FL, USA) and converted to Ω × cm^2^ based on the area of transwell inserts (0.33 cm^2^). 

Transport studies were performed based on our previously published method [9], and Hank’s balanced salt solution (HBSS) was supplemented with 25 mM HEPES, 3.6 mg/mL BSA, 0.5 mM taurocholate, and 2 mM oleic acid. Blank HBSS (200 μL) was added to the apical side of the trans-well, and the cell monolayers were incubated for 1 h at 37 °C. Next, 1 mL supplemented HBSS was added to the basolateral compartment of the transwell plate. To investigate the influence of inhibiting SR-B1 on the transport, 10 μM BLT-1 dissolved in HBSS (2.0 mM BLT-1 in DMSO diluted 200 times in HBSS) was added and pre-incubated for 1 h at 37 °C. Subsequently, the medium from the apical side of the trans-well plate was removed and the donor solution was added (200 μL of mixed micelles suspended in blank HBSS, containing 350 μM vitamin K and 1.58 mM lipid, 500 μg/mL FITC-dextran (20 kDa) as control group). At different time points (30, 60, 90, 120, 150 and 180 min), 500 μL medium samples from the basolateral side of the transwell plate were withdrawn and replaced by the same volume of above mentioned supplemented HBSS.

To determine the vitamin K concentrations, 100 μL samples of the basolateral medium was mixed with 200 μL ethanol [9]. After vortexing for 10 min, the samples were centrifuged for 10 min at 8000 *g* at room temperature. Next, 100 μL supernatant of the different samples was injected onto an HPLC column. The HPLC method was based on our previously described method [9]. The mobile phase consisted of 880 mL methanol, 120 mL ACN, 1.2 g ZnAc, 10 mL HAc, and 5 mL water, which was flushed with nitrogen gas for 5 min before use. A reduction column (5.5 cm × 1.5 cm, filled with zinc powder) was attached to a SunFire C18 column to convert vitamin K into its reduced form for fluorescence detection. The flow rate was 1.2 mL/min, the column temperature was 30 °C, and the excitation and emission wavelengths were 246 and 430 nm, respectively. The vitamin K concentrations in the different samples were calculated using a calibration curve of vitamin K (linear from 0.024–50 ng/mL).

The apparent permeability coefficients (P_app_) of vitamin-K-loaded mixed micelles was calculated using the following equation: (3)$$P_{\mathrm{app}}(\mathrm{cm}/s)=\frac{\Delta Q}{\Delta t}\times \frac{1}{A}\times C_{0}$$
where ΔQ/Δt is the amount of vitamin K transported across the cell layer and accumulates in the basolateral compartment per second, A is the surface area of the insert (0.33 cm^2^), and C_0_ (0.158 mg/mL) is the vitamin K concentration of the donor solution added to the apical compartment.

### 2.5. Uptake of Rhodamine-Labeled Mixed Micelles and Colocalization with SR-B1 as Studied by Confocal Microscopic Analysis

Caco-2 cells were seeded in 24-well plates at a density of 2.5 × 10^4^ per well and cultured for 21 days at 37 °C with 5% CO_2_. The medium for differentiated Caco-2 cells is described in Section 2.3 and was refreshed every 2 days. Next, 500 µL rhodamine-labeled non-PEGylated, 30%-PEGylated or 50%-PEGylated mixed micelles (lipid concentrations were 1.58 mM) with 350 µM vitamin K diluted in blank DMEM were added and incubated for 2 h at 37 °C. Subsequently, the medium was removed and the cells were washed two times with PBS. To investigate the influence of inhibition of SR-B1 on the uptake of vitamin-K-loaded mixed micelles, the cells were pre-incubated with 500 µL medium with 10 µM block lipid transport-1(BLT-1) at 37 °C for 1 h. Subsequently, 500 µL rhodamine-labeled non-PEGylated, 30%-PEGylated or 50%-PEGylated mixed micelles (lipid concentrations were 1.58 mM), with 350 µM vitamin K diluted in blank DMEM, were added and incubated for 2 h at 37 °C. Next, the medium was removed and the cells were washed two times with PBS. Subsequently, the cells were firstly fixed with 4% paraformaldehyde for 10 min at room temperature and then washed with PBS. Next, the cells were permeabilized with 0.05% saponin for 2 min and washed with PBS. Cells were subsequently incubated with 3% BSA solution in PBS for 1h at room temperature before incubation with the primary antibody against SR-B1 (200 µL, mouse anti human CLA-1, 1:100 dilution in PBS with 3% BSA). After 1 h incubation at room temperature, the cells were washed with PBS 3 times and subsequently incubated with Alexa 488-labeled goat anti-mouse IgG (1:400 dilution in 3% BSA solution, 200 µL) for 1 h at room temperature. Next, cells were incubated with Fluoroshield™ solution containing DAPI for nucleus staining. Images were acquired using a confocal microscope (Yokogawa CV 7000s, Yokogawa Electric Corporation, Tokyo, Japan) with 60× objective. The excitation wavelengths were 405, 488 and 561 nm, while the corresponding acquisition channels were 445 ± 45, 525 ± 50, and 600 ± 37 nm, respectively. Image J was used to analyze mean fluorescence and calculate the Pearson’s correlation coefficients (R value) for quantification of the degree of colocalization between SR-B1 and vitamin-K-loaded mixed micelles.

### 2.6. Binding and Uptake of Mixed Micelles by Hela Cells Overexpressing SR-B1 Studies by Confocal Microscopic Analysis

The binding to and uptake of mixed micelles by transfected Hela cells were imaged by confocal microscope. Hela cells were seeded in a 96-well plate at a density of 10,000 per well and incubated overnight at 37 °C. The cells were transfected with a plasmid containing SR-B1 cDNA with C-terminal GFP tag using Lipofectamine 3000 according to the protocol provided by the manufacturer. In short, 20 µL of the lipofectamine/cDNA dispersion containing 0.1 µg cDNA, 0.9 µL lipofectamine 3000 and 0.2 µL P3000 reagent was added to each well and incubated for 48 h at 37 °C, to obtain transfected Hela cells. The medium is described in Section 2.3. The transfected Hela cells were incubated with 200 µL rhodamine-labeled non-PEGylated, 30%-PEGylated or 50%-PEGylated mixed micelles (lipid concentration was fixed at 1.58 mM) with or without vitamin K (350 µM) at either 4 or 37 °C for 2 h in blank DMEM, respectively. Next, the medium was removed and the cells were washed 3 times with PBS. Subsequently, the cells were fixed with 4% paraformaldehyde for 10 min at room temperature, permeabilized with 0.05% saponin for 2 min, and then washed with PBS 3 times. The cells were finally incubated with Fluoroshield™ solution containing DAPI for nucleus staining. Images were acquired using a confocal microscope as described in Section 2.5. The expression of SR-B1 by the transfected Hela cells was also measured by immunofluorescence staining, as described in Section 2.5. Briefly, the cells were fixed with 4% paraformaldehyde and permeabilized with 0.05% saponin. Next, the cells were incubated with a primary antibody against SR-B1 and Alexa 555-labeled goat anti-mouse IgG and then exposed to the Fluoroshield™ solution containing DAPI. Images were acquired as described in Section 2.5.

### 2.7. Binding and Uptake of Mixed Micelles by Hela Cells Overexpressing SR-B1 Measured by Fluorescence-Activated Cell Sorting (FACS) 

Hela cells were seeded in a 24-well plate at a density of 100,000 per well and incubated overnight at 37 °C. Next, the cells were transfected using same method as described in Section 2.6 with modifications. Briefly, 50 µL of the lipofectamine/cDNA dispersions containing 4.5 µL lipofectamine 3000, 0.25 µg cDNA and 1 µL P3000 reagent was added to each well. Expression of SR-B1 by the transfected and non-transfected Hela cells was measured by immunofluorescence staining. In detail, the cells were detached after addition of 200 µL EDTA/trypsin solution at 37 °C for 5 min. Next, 400 µL medium was added and the cells were pelleted by centrifuging for 5 min at 400 *g*. After washing with PBS 3 times, the collected cells were firstly fixed with 4% paraformaldehyde for 10 min at room temperature and then washed with PBS. Next, the cells were incubated with 3% BSA solution in PBS for 1 h at room temperature before incubation with the primary antibody against SR-B1 (200 µL, mouse anti human CLA-1, 1:100 dilution in PBS with 3% BSA). After 1h incubation at room temperature, the cells were washed 3 times with PBS and subsequently incubated with Alexa 555-labeled goat anti-mouse IgG (1:400 dilution in 3% BSA solution, 200 µL) for 1h at room temperature. Next, excess antibody was removed by washing the cells 3 times with PBS, and the cells were suspended in 200 µL PBS. The cells were finally analyzed by FACS (LSR Fortessa, BD Bioscience, San Jose, CA, USA) using laser lights of 488 nm and 561 nm for detection of GFP and Alexa 555, with detection at 530 ± 30 nm and 610 ± 20 nm, respectively. The FACS data of 10,000 cells were analyzed using Flowlogic software V.8.

To study binding and uptake of mixed micelles, the transfected Hela cells were incubated with 500 µL rhodamine-labeled non-PEGylated 30%-PEGylated or 50%-PEGylated mixed micelles (lipid concentration was 1.58 mM) with or without vitamin K (350 µM) loading at either 4 or 37 °C for 2 h in blank DMEM, respectively. Next, the medium was removed and the cells were 3 times washed with PBS. Subsequently, the cells were detached after addition of 200 µL EDTA/trypsin solution at 37 °C and incubation for 5 min. Next, 400 µL medium was added and the cells were pelleted by centrifugation for 5 min at 400 *g*. After washing with PBS 3 times, the Hela cells were suspended in 200 µL PBS and subsequently analyzed by FACS with the same wavelength settings as described above for detection of SRB1/GFP and rhodamine. 

### 2.8. Surface Plasmon Resonance (SPR) to Study the Interaction between SR-B1 and Mixed Micelles of Different Compositions

Surface plasmon resonance (SPR) studies were performed using a MX96 SPR instrument (IBIS Technologies, Enschede, The Netherlands). Recombinant extracellular domain of SR-B1 was immobilized on a SPR sensor chip covered with a dextran layer that contained carboxylic acid groups (P–COOH type, SensEye, Enschede, The Netherlands), following the standard amine coupling protocol provided by the manufacturer. In brief, carboxyl groups in the dextran layer on the sensor chip were activated for 10 min using 0.4 M EDC/0.1 M NHS (1:1 (v:v)) in activation buffer (50 mM MES, 0.005% Tween 20, pH 5.4) at 25 °C. To identify optimal protein coupling conditions, recombinant SR-B1 (concentrations of 2, 4 and 8 µg/mL) in PBS (0.2 M KH_2_PO_4_), and buffers of four different pHs (4.0, 4.5, 5.0 and 5.5, adjusted with 0.2 M NaOH and 1 M HCl), were prepared, and the resulting solutions were deposited using a CFM Printer (IBIS Technologies, Enschede, The Netherlands) at different spots on the chip for 1 h at room temperature. Subsequently, unreacted activated carboxyl groups were deactivated by 100 nM ethanolamine hydrochloride (pH 8.0) for 10 min at 25 °C. The SR-B1 spot (pH 4.0, 8 μg/mL) with RLL (response of local ligand) value of 2452 ± 16 Ru (1 Ru = 1 pg/mm^2^) [20] was chosen as representative to investigate binding and competition studies. 

Mixed micellar dispersions were diluted in SPR running buffer (10 mM HEPES, 150 mM NaCl and pH 7.4) and flown over the coated sensor chip for 20 min. Non-PEGylated mixed micelles with different vitamin K loadings (empty non-PEG, 5VK-nonPEG, 10VK-nonPEG, or 20VK-nonPEG), PEGylated mixed micelles with different PEG density (containing 50%, 30% and 15% DSPE-PEG) with and without 8 µM vitamin K, or DSPE-PEG-only micelles without vitamin K were injected into the SPR cell, all at the same concentration of total lipids (36 mM). To obtain the dissociation constants (K_D_), mixed micelle dispersions of different dilutions (with lipid concentrations for 20VK-nonPEG: 1.3–46.1 µM, 20VK-30%PEG: 2.8–900 µM, 20VK-50%PEG: 4–1781 µM, Empty non-PEG: 27.8–711.7 µM, Empty-30%PEG: 12.3–1067.6 µM, Empty-50%PEG: 8.2–1601 µM) were injected and the responses were recorded. The micelles hardly desorbed from the recombinant SR-B1 coated chip in plain running buffer. Therefore, 0.05% Trion X100 was used as regeneration buffer and run for 60 **s** over the coated chip followed by a 2 min dissociation step with SPR running buffer, which was repeated 5 times. Non-PEGylated micelles were injected (8 μM vitamin K, 36 μM lipid) regularly during the binding and regeneration process to check for possible changes in the binding capacity of receptor. Control experiments showed that this treatment neither caused protein detachment nor reduction in binding capacity of SR-B1. All measurements were carried out at 25 °C. The data were analyzed using Sprint X (IBIS Technologies, Enschede, the Netherlands) and GraphPad Prism software [21]. Response units based on the averaged response signal at equilibrium, i.e., between 1100 and 1200 s of the association phase (Ru, y-axis), were plotted against lipid concentration of the micelles (x-axis). One site-specific saturation model was used to calculate dissociation constant at equilibrium (K_D_). 

A primary IgG antibody against SR-B1 was used to study its competition with vitamin-K-loaded micelles for binding to SR-B1. The antibody, at either 0.5 or 25 μg/mL in running buffer, was injected before or after binding of mixed micelles (20VK-nonPEG, 20VK-30%PEG and 20VK-50%PEG, with 8 μM vitamin K and 36 μM lipid).

### 2.9. Molecular Docking to Study the Binding Sites of SR-B1 with Vitamin K and PEG

The structures of vitamin K and pentamer of ethylene glycol were generated in ChemDraw 19.0 and subsequently imported in Chem3D 19.0 and saved as mol2 file. From this starting point, a library of conformers was generated using OMEGA2 software (Release 4.0.0.4, OpenEye Scientific Software, Inc., Santa Fe, NM, USA; www.eyesopen.com, accessed on 19 October 2021) [22] using default settings, which was limited to 200 conformers. The SR-B1 protein conformation (Q8WTV0, Appendix A) was generated from the homology model based on the template of Lysosome membrane protein 2 (LIMP-2, Template 6i2k.3.T, Appendix A) from SWISS MODEL, with 36% sequence identity [23,24]. This model was the input for MAKE RECEPTOR (Release 3.5.0.4, OpenEye Scientific Software, Inc., Santa Fe, NM, USA; www.eyesopen.com). For “Cavity detection” slow and effective “Molecular” method was used for detection of binding sites. Inner and outer contours of the grid box were also calculated automatically using “Balanced” settings for “Site Shape Potential” calculation. Fred optical engineering software (FRED, Release 3.5.0.4, OpenEye Scientific Software, Inc., Santa Fe, NM, USA; www.eyesopen.com) was used for docking [25,26,27]. A hit list of top 100 ranked molecules was retrieved, and the best-ranked FRED-calculated pose was inspected visually and used for analysis and representation. Results were evaluated in visualization software VIDA 4.4.0 (OpenEye Scientific Software, Inc., Santa Fe, NM, USA; http://www.eyesopen.com).

### 2.10. Statistical Analysis

Statistical analysis was done by GraphPad Prism software Version. 6.01. Statistical significance was performed by two-tail unpaired *t*-test and two-way analysis of variance (ANOVA). A value of *p* < 0.05 was considered significant. Statistical significance is depicted as * *p* ˂ 0.05. 

## 3. Results

### 3.1. Preparation and Characterization of Vitamin-K-Loaded Mixed Micelles 

Mixed micelles based on lipids of different compositions with and without vitamin K loading (varying PEG density and vitamin K loading) were prepared by a film hydration method followed by membrane extrusion [5]. Table 1 shows encapsulation efficiencies of more than 90%, with corresponding loading % as shown in Table 2. The size and zeta potential of the extruded micelles were analyzed by DLS and Zetasizer (Table 1, Appendix A) [28]. In line with previous observations [5], the average size of empty and vitamin K-loaded micelles increased from ~7 to 11 nm with increasing PEG content, which can be explained by the thickness of the PEG 2000 shell of 1.5–3.5 nm [29]. The polydispersity index (PDI) of the different micellar dispersions was < 0.2, demonstrating a relatively narrow size distribution. Table 1 also shows that with increasing PEG content, the zeta potential at pH 7.4 became less negative from −20.6 ± 2.2 mV to −5.1 ± 0.2 mV, which can be ascribed to shielding of the negatively carboxylate anions of glycocholic acid by the PEG chains [30,31].

### 3.2. Transport of Vitamin-K-Loaded Mixed Micelles through Differentiated Caco-2 Monolayers

The Caco-2 cell line, derived from human colorectal adenocarcinoma, has been widely used for studying transport kinetics of drugs and drug-loaded nanoparticles [32,33]. This cell line can spontaneously differentiate to resemble a small intestine-like phenotype with enterocyte-like absorptive properties [34]. In the present study, differentiated Caco-2 cell monolayers were therefore used to investigate the transport of vitamin-K-loaded mixed micelles. Typical TEER values of monolayers reached 1598 ± 150 Ω × cm^2^ after 21 days of cell culture, demonstrating that the cultured monolayers were confluent (Appendix A) [35]. In support, FITC-dextran added to the apical side was not detected in the basolateral part after 180 min of incubation (lower than limit of detection 0.15 μg/mL), confirming the formation of an intact monolayer with tight junctions. The permeability coefficients of vitamin K solubilized by the different micelles through differentiated Caco-2 monolayers were determined by the linear slope of the plots of cumulative amounts of transported vitamin K against time (Appendix A). Based on equation 3, the calculated transport permeability coefficients (see Figure 1) were (5.4 ± 0.4) × 10^−10^, (3.6 ± 0.5) × 10^−10^, and (3.8 ± 1.0) ×10^−10^ cm/s for vitamin-K-loaded mixed micelles with 0%, 30% and 50% PEGylation, respectively. When the differentiated Caco-2 cells were incubated with 10 μM BLT-1 (i.e., inhibitor for the SR-B1) [9], 30–50% decrease in the transport permeability coefficients value was observed ((3.8 ± 0.4) × 10^−10^, (2.4 ± 1.4) × 10^−10^, (1.8 ± 1.0) × 10^−10^ cm/s, respectively). The lower permeability of the Caco-2 monolayer for mixed micelles with and without PEGylation upon inhibition of SR-B1 strongly suggests a role for SR-B1 in transport of either intact micelles or vitamin K released from the micelles and urged us to further investigate the possible interactions between micelles and SR-B1.

### 3.3. Colocalization Studies of SR-B1 and Vitamin-K-Loaded Mixed Micelles

We applied immunofluorescent staining of the receptor and studied the co-localization with rhodamine-PE-labeled micelles. To this end, Caco-2 cell monolayers were used that had been in culture for 21 days, at which time expression of SR-B1 was highest. SR-B1 was detected upon incubation with a matching antibody-labeled with a green fluorescent probe, turning yellow when merged with the red fluorescence of rhodamine-labeled mixed micelles. As shown in Figure 2A, the red fluorescence of the micelles colocalized with the green fluorescence of the receptor after 2 h incubation at 37 °C, with the rhodamine fluorescence and the intensity of the yellow color being more pronounced for the non-PEGylated than for the 30 and 50%-PEGylated micelles (1st, 3rd and 5th row in Figure 2A, respectively). 

BLT-1 is a synthetic inhibitor of SR-B1 [36], of which its thiosemicarbazone moiety binds to the free thiol of cysteine 384 in the extracellular domain of the receptor [37], which is important for its structural integrity and function. Cellular uptake of rhodamine-labeled micelles decreased substantially after pre-incubation of the Caco-2 cells with BLT-1 (Figure 2A, the 2nd, 4th and 6th row), i.e., by 37 ± 12%, 64 ± 9% and 55 ± 21% (Figure 2B). Pearson’s coefficients (R value) as a measure of the colocalization ratio of mixed micelles and SR-B1 are presented in Figure 2C. Inhibition of the receptor by BLT-1 was accompanied by a decrease in the Pearson’s coefficients (for non-PEGylated micelles from 0.63 ± 0.05 to 0.53 ± 0.03, 30%-PEGylated micelles from 0.47 ± 0.02 to 0.39 ± 0.03 and 50%-PEGylated micelles from 0.34 ± 0.01 to 0.24 ± 0.06), demonstrating that the uptake pathway of mixed micelles was blocked by inhibition of the scavenger receptor B1 with BLT-1. These data obtained through labeling of the micelles by fluorescence further confirm the role of SR-B1 in the uptake of vitamin-K-loaded mixed micelles, both with and without PEGylation, most likely through binding of intact micelles rather than released vitamin K. 

### 3.4. Binding and Uptake of Mixed Micelles by Hela Cells Overexpressing SR-B1/GFP Revealed by Confocal Microscopy

To investigate the binding and uptake of mixed micelles mediated by SR-B1, Hela cells were selected and transfected to overexpress SR-B1. Native Hela cells hardly express SR-B1 [38,39], and thus induction of expression by transfection minimizes the influence of endogeneous SR-B1 and other intestinal transporters that are otherwise present in Caco-2 cells [40]. After transfection with a plasmid encoding for the fusion protein of SR-B1 and GFP, green fluorescence from GFP was observed and colocalized with SR-B1-labeled through immunofluorescence staining (red fluorescence), as shown in Appendix A, demonstrating successful transfection of the plasmid. 

At 4 °C (Figure 3A), the internalization process of micelles was inhibited and micelles showed mainly association with the membrane of the cells. Especially, vitamin-K-loaded non-PEG micelles (20VK-nonPEG) showed extensive association with Hela cells that displayed GFP fluorescence (i.e., overexpressing SR-B1/GFP), as shown in Figure 3A. For micelles with 30% and 50% PEGylation, there was less red fluorescence associated with Hela cells showing the SR-B1/GFP signal, as compared with cells incubated with non-PEGylated micelles, but the fluorescence was still stronger than cells without SR-B1/GFP signal. Hela cells incubated with micelles at 37 °C (Figure 3B) showed almost the same trend as the cells incubated at 4 °C. The presence (left panels) or absence of vitamin K in the micelles (right panels) did not have an effect on the binding and uptake of the mixed micelles, regardless the degree of PEGylation. Cells not overexpressing the SR-B1/GFP hardly showed binding and internalization of the micelles (Figure 3A,B). On the other hand, overexpression of SR-B1/GFP among cells resulted in much higher binding (4 °C, Figure 3A) and internalization (37 °C, Figure 3B) of the mixed micelles, strongly demonstrating that SR-B1 mediates the endocytosis of the mixed micelles with and without PEGylation.

### 3.5. FACS Analysis of Binding and Uptake of Mixed Micelles by Hela Cells Overexpressing SR-B1 

FACS was used to quantify overexpression of SR-B1/GFP by Hela cells, as well as binding and uptake of mixed micelles by these transfected cells (cells exposed to transfection reagents, Figure 4B,C) and non-transfected Hela cells (cells not exposed to transfection reagents, Appendix A). Based on GFP green fluorescence at the C-terminus of SR-B1, around 20–30% of the Hela cells were transfected successfully (Figure 4A and Appendix A). SR-B1/GFP was labeled through immunofluorescence staining by Alexa 555 conjugated antibody (Appendix A), resulting in an increase from 7.4 to 32% Hela cells showing Alexa 555 signal upon transfection. In the histogram (Appendix A, right), the population of transfected Hela cells showing Alexa 555 fluorescence (red curve, Appendix A, bottom right) together with a SR-B1/GFP signal was relatively large compared with the number of Hela cells without SR-B1/GFP signal (blue curve, Appendix A, bottom right) in the same transfected well, and compared to control cells without transfection (Appendix A, top right). Accordingly, the expression of SR-B1 is positively correlated with SR-B1/GFP intensity observed in the dot plots (Appendix A, bottom middle), indicating higher expression of SR-B1 in Hela cells with stronger SR-B1/GFP signal. This is expected because the cells were transfected with a SR-B1/GFP fusion protein cDNA. 

To investigate the cellular binding and internalization of mixed micelles, non-transfected and transfected Hela cells were incubated with rhodamine-labeled micelles with and without vitamin K loading at 4 and 37 °C, respectively, and subsequently analyzed by FACS. The dot plots are divided into four regions: unstained cells are distributed in R4 (region 4), R1 includes cells that express SR-B1/GFP, R2 contains cells that show both SR-B1/GFP and micelles (rhodamine) fluorescence, and R3 contains cells that only show rhodamine fluorescence. Binding and uptake of rhodamine-labeled micelles was detected as a right shift of the population on the X axis. For non-transfected Hela cells that do not show SR-B1/GFP fluorescence, the population partly shifts from R4 to R3 (Appendix A). From the normalized median fluorescence shown in Appendix A, it is noted that rhodamine fluorescence decreased with increasing PEG density upon incubation of non-transfected Hela cells with non, 30% and 50% PEGylated mixed micelles with and without vitamin K at both 4 and 37 °C, indicating reduced binding and uptake, respectively. 

In transfected cells (Figure 4B,C), the cell dots in R1 shift to R2 to different extent upon incubation of rhodamine-labeled micelles. For non-PEGylated micelles with and without vitamin K (first row of Figure 4B,C), the binding (Figure 4B) and uptake (Figure 4C) were confirmed by significant rhodamine fluorescence intensity for Hela cells with GFP signal (R2), with an almost linear correlation between rhodamine and SR-B1/GFP fluorescence intensity: the higher the SR-B1/GFP fluorescence (i.e., SR-B1 expression), the higher the micelle binding and uptake. For the 30% and 50%-PEGylated micelles with and without vitamin K, the difference of rhodamine fluorescence in cells with and without GFP signal is not significant but still shows an increasing trend along with the SR-B1/GFP fluorescence (Figure 4B, second row and third row). 

The quantified median fluorescence intensity (MFI, Figure 4D,E) indicates that the binding and uptake of non-PEGylated micelles are more significantly dependent on overexpression of SR-B1/GFP than was observed for PEGylated mixed micelles. Cell associations (Figure 4D) of non-PEGylated micelles with Hela cells with SR-B1/GFP signal (R1 + R2) were 2.8-fold (with vitamin K) and 3.8-fold (without vitamin K) higher than that in Hela cells without SR-B1/GFP signal (R3 + R4). This increase was even more significant with cells incubated with non-PEGylated micelles with and without vitamin K at 37 °C, showing 3.5 and 5.1-fold enhancement of the MFI, respectively (Figure 4E). It is worth noting that non-PEGylated mixed micelles with vitamin K loading showed higher association with and internalization by Hela cells (with and without overexpression of SR-B1/GFP) than similar non-PEGylated micelles but without vitamin K loading. On the other hand, the presence or absence of vitamin K did not affect the binding and uptake for 30% and 50% PEGylated mixed micelles, indicating that vitamin K did not play a role in uptake and binding of these PEGylated micelles. Nevertheless, Hela cells with overexpression of SR-B1/GFP and incubated with 30% and 50%-PEGylated micelles showed 1.5- to 1.8-fold enhancement of association and internalization compared to cells without SR-B1/GFP, respectively, indicating that SR-B1 still plays a role in the binding and uptake of the PEGylated micelles. Combining with the data from Section 3.3 and Section 3.4, it is concluded that the SR-B1 has affinity for mixed micelles and mediates the endocytosis of vitamin-K-loaded mixed micelles with and without PEGylation.

### 3.6. SPR Analysis to Study the Affinity between SR-B1 and Mixed Micelles

Surface plasmon resonance (SPR) as a label-free, real-time, and sensitive method is a frequently applied technique to examine molecular interactions, e.g., those between a receptor and a ligand [41]. The recombinant extracellular domain of SR-B1 was immobilized on the SPR chip using standard amine coupling method [42]. The immobilization of the SR-B1 was optimized by varying the protein concentration and pH during coupling (Appendix A) with 1 h reaction at room temperature. The condition (8 μg/mL at pH 4.0) resulting in the highest RLL 2452 ± 16 Ru and thus protein density was selected to study the interaction with of micelles of different compositions and at different concentrations. Appendix A shows typical sensorgrams of real-time interaction between micelles and SR-B1. The one site-specific saturation model was used to calculate K_D_ and B_max_ values (Table 2), as described in the Section 2. Subsequently, the saturation curve was normalized to the response signal observed upon association of SR-B1with the micelles at the highest concentration (Figure 5B).

Figure 5A (left) shows the SPR response after binding of non-PEGylated micelles at a fixed lipid concentration of 36 µM and different vitamin K loadings to the immobilized SR-B1 protein. This figure shows that the SPR signal increases 10-fold, when vitamin K loading increases from 2 to 8 μM (corresponding to 5VK-nonPEG and 20VK-nonPEG). The dissociation constant of micelles with the highest vitamin K loading (K_D_ of 9.2 ± 0.1 μM) is 17-fold lower than that of the corresponding empty micelles (K_D_ of 159 ± 19 μM) (Table 2). 

The data of Figure 5A and Table 2 show that the binding strength of the PEGylated micelles to SR-B1 decreases with increasing PEG density. Eventually, neutral micelles composed only of DSPE-PEG had no affinity for the SR-B1 receptor (Figure 5A). However, in contrast to non-PEGylated micelles, the binding strength is not dependent on vitamin K loading content. Remarkably, 15% PEGylated micelles interact with SR-B1, even without vitamin K (Figure 5A, right panel). Table 2 shows that the K_D_ for the 30%-PEGylated micelles with and without vitamin K loading is similar (59 ± 9 μM and 37 ± 7 μM, respectively), as well as the K_D_ values for the 50%-PEGylated micelles with and without vitamin K loading (133 ± 34 μM and 128 ± 25 μM, respectively). The obtained data demonstrate that interaction of PEGylated micelles with SR-B1 is mediated by PEG, and vitamin K does not contribute to this interaction, possibly due to shielding of the compound present in the core of the micelles by the PEG corona. Table 2 and Figure 5 also show that for micelles without vitamin K loading, the 30%-PEGylated micelles have the highest affinity, i.e., the lowest K_D_ value (37 ± 7 μM), whereas both non-PEGylated empty micelles (159 ± 19 μM) and 50%-PEGylated empty micelles (128 ± 25 μM) have a lower affinity. This suggests that the PEG density has an effect on the affinity, possibly as a result of a different PEG conformation. The relationship between different topographical structures of PEG and their affinity for proteins has been investigated before [43,44]. The conformation of surface grafted PEG is dependent on both its molecular weight and nanoparticle density [45]. The mushroom conformation occurs when the average distance between the attachment points of two adjacent PEG chains (*D*) is greater than Flory dimension (*R_f_*) of the polymer. On the other hand, for the brush conformation, PEG chains are grafted closer together, forcing the polymers chains to take an elongated conformation, against their natural tendency to coil [46,47]. The surface conformation of PEG on the micelles was calculated based on the ratio of the Flory dimension (*R_f_*) to the average distance between adjacent PEG chains (*D*), using the following Equations (4)–(6) [46]:(4)$$D={\left(A\right)}^{\frac{1}{2}}$$
(5)$$A=\frac{M\times 4\pi {\left(\frac{d_{0}}{2}\right)}^{2}}{\mathrm{wt}\%\mathrm{PEG}\times \rho \times \frac{4\pi }{3}{\left(\frac{d_{\mathrm{DLS}}}{2}\right)}^{3}\times N_{A}}$$
in which *A* is the average area occupied by one PEG chain; d_DLS_ is the diameter of micelles measured by DLS; d_0_ is diameter of the core of the micelles, equal to diameter of non-PEGylated mixed micelles; M is the molecular weight of PEG (2000 g/mol); N_A_ is Avogadro’s number; wt% PEG is PEG weight percentage of the micelles; and ρ is the average density of mixed micelles (1.25 g/cm^3^) [48]. The Flory radius (end-to-end distance) of PEG 2000 is 3.44 nm, calculated using the following equation:(6)$$R_{f}\approx N^{3/5}a$$
in which *N* is the degree of polymerization (*N* = 45 for PEG 2000) and *a* is the monomer size, (*a* = 0.35 nm for CH_2_–CH_2_–O) [49]. Calculated *R_f_*/*D* values below 1.0 indicate a mushroom conformation, while those above 1.0 indicate brush conformation [50]. As shown in Appendix A, when the PEG density increases from 15 to 50%, *Rf*/*D* increases from 1.20 to 2.56, indicating more brush-like conformation with increasing PEG density. Therefore, the decreasing affinity of the PEGylated mixed micelles for the SR-B1 receptor with increasing PEG density can be explained by the change in the PEG conformations, with lowest K_D_ for the empty micelles that were surface decorated with PEG in a relatively low-density brush-like conformation (i.e., in this case with 30% PEGylation, see Figure 5B). Correspondingly, in the recent study of Raith et al., it is shown that PEG-PBD spheres with relatively high-density brush-like conformation do not interact strongly with SR-B1, as opposed to PEG on the surface of a more flexible entity like filomicelles [17]. 

Appendix A shows that B_max_ values of non-, 30% and 50% PEGylated mixed micelles reach similar levels at saturation concentrations (i.e., Ru of around 150 for empty micelles and Ru of around 400 for vitamin-K-loaded micelles), independent of PEG content. SPR detects refractive index changes, which is related to mass changes in the vicinity of the interface, caused by ligand binding to the immobilized receptor. This indicates that similar masses are bound to the receptor, most likely complete micelles in all cases, rather than individual molecular components such as vitamin K. In fact, B_max_ values of vitamin-K loaded micelles are 2.4-, 2.2- and 1.4-fold higher than empty micelles for non, 30%, and 50% PEGylated mixed micelles, respectively. Such higher values for vitamin-K-loaded micelles (by 4%–6.9% weight percentage of whole micelle mass) are probably caused by a higher refractive index for vitamin-K-loaded mixed micelles with respect to the empty micelles [20].

Taken together, it is concluded from SPR data that both vitamin K and PEG bind to the SR-B1. As mentioned, the micelles without PEG-decoration may come in close contact with the receptor and expose vitamin K on their surface, subsequently leading to binding of the micelles to the receptor. Our data show that PEGylation prevents vitamin K from interacting with SR-B1, because the vitamin-K-loaded and empty micelles have similar K_D_ values when PEGylated. On the other hand, PEG chains present in the shell of the micelles appear to interact with SR-B1, dependent on their conformation. In agreement with the SPR data, it is indeed concluded from the transport (Figure 1) and colocalization studies (Figure 2) that internalization of vitamin-K-loaded mixed micelles is mediated by SR-B1, both with and without PEGylation. Although PEG resulted in less interaction with SR-B1 and reduced endocytosis by Hela cells (Figure 3 and Figure 4) and lower internalization by and subsequent transport through Caco-2 cells, omission of vitamin K from the PEGylated micelles did not show an effect on the micelle binding and uptake by Hela cells overexpressing SR-B1. Additionally, in agreement with SPR data, empty non-PEGylated micelles showed less binding to and internalization by Hela cells overexpressing SR-B1 than corresponding micelles loaded by vitamin K (Figure 4C,D). 

However, in Hela cells overexpressing SR-B1, empty non-PEGylated micelles show stronger binding and uptake than empty PEGylated micelles, which is different from the trend observed in SPR data that showed the highest K_D_ (lowest affinity) for the empty non-PEGylated micelles (Table 2). Possibly, those empty non-PEGylated micelles are taken up via a different mechanism than mediated by SR-B1. Finally, it must be noted that the physiological environment is much more complicated than the SPR system using only a receptor and a well-defined buffer. Furthermore, the receptor-mediated endocytosis process involves complex signaling pathways and energy transfer, which are impossible to simulate using protein immobilized on a chip [51].

To get further mechanistic insight into the interaction between mixed micelles and SR-B1, we performed a competition study using an antibody against SR-B1 that binds to the amino acid sequence 104–294 of the receptor, based on the information provided by the manufacturer [52]. It is noted that the diameters of the antibody and micelles are both around 10 nm. Injection of antibody at concentrations of 0.5 μg/mL and 25 μg/mL generates responses of 25 ± 3 Ru (Appendix A) and of 1432 ± 86 Ru (Appendix A), respectively. The subsequent washing with regeneration buffer completely removed the adsorbed antibody at relatively low concentration (0.5 μg/mL). As shown in Appendix A, most of the receptor-bound 20VK-nonPEG micelles remained associated upon exposure to the clean running buffer (i.e., with Ru of approx. 361 ± 25) and could only be removed by the regeneration solution containing 0.05% Trion X100. Without regeneration, i.e., with the micelles still bound to the receptor, subsequent injection of the antibody caused an additional response of 25 ± 3 Ru, i.e., similar to the response of interaction of the antibody with fully regenerated SR-B1 receptor (Appendix A). In addition, after first injecting the antibody and then the 20VK-nonPEG micelles without in between regeneration (i.e., third time 20VK-nonPEG was injected in Appendix A), again an increase of 360 ± 23 Ru was observed that was around equal to the signal detected without previous exposure to the antibody. These results indicate that the antibody at low concentration and 20VK-nonPEG do not compete with each other in binding with SR-B1. Appendix A shows that when the chip was exposed to a solution of antibody of relatively high concentration (25 μg/mL), the antibody could not be fully desorbed upon washing with regeneration buffer (response maintaining at 583 ± 14 Ru). This phenomenon was used to check recovery of the micelles before and after the SR-B1 receptor was (partly) blocked with antibody. Formulations (20VK-nonPEG, 20VK-30%PEG and 20VK-50%PEG) were found to have similar response before and after SR-B1 blocking by the antibody. It demonstrated that SR-B1 could bind with vitamin-K-loaded mixed micelles and antibody and there is probably no competition between both. 

### 3.7. Molecular Docking to Study the Binding Sites of SR-B1 with Vitamin K and PEG 

Docking simulation was examined to study the possible binding site of vitamin K and PEG for SR-B1. One of the binding poses of vitamin K binding to SR-B1 is shown in Figure 6. Numerous lipophilic amino acid residues are close to the ligand as depicted and include: Leu140, Met147, Phe199, Phe201, Leu211, Trp237, Arg335 and Phe336, and other related low-energy poses, including interaction with Phe195, Phe198, Phe199, Leu155, and Ser214. Notably, the predicted binding location of cholesterol is essentially the same as for vitamin K, and many contacts with the mentioned amino acid residues are also made by the predicted ensemble of cholesterol poses. Furthermore, the calculated Chemgauss4 scores, which are a measure for the expected affinity in FRED [53], are slightly in favor of cholesterol (−10.7 for vitamin K and −11.8 for cholesterol). To predict whether PEG would likely be bound to the site of cholesterol/vitamin K as well, a pentamer of ethylene glycol was used for docking to SR-B1. This time, the molecules occupied two positions of the lipophilic channel, one of which was similar to the mentioned positions of vitamin K and cholesterol, while the second occupied the entrance of the tunnel with notable interactions with Phe245, Trp246, Cys280 and Cys334. The best pose had a significant Chemgauss4 score of −7.6, which, in combination with the inspection of the binding poses, makes binding of PEG to the receptor a real possibility. Nevertheless, the score was lower than the other mentioned ligands, likely due to its lower lipophilicity, higher desolvation penalties, and more flexible structure. As revealed by docking simulation, the binding sites for PEG and vitamin K were different from the binding site of the antibody investigated in our studies that occupies amino acid sequence 104–294, which explains why binding of micelles and antibody was not competitive (see Section 3.6). SR-B1 forms oligomers that mediate lipid uptake, with involvement of *N*-terminal transmembrane glycine motif (Gly15_Gly18_Gly25) [54]. Moreover, four cysteines (Cys280, Cys321, Cys323 and Cys334) participating in the formation of two intramolecular disulfide bonds were shown to be involved in forming a tunnel cavity through oligomerization of SR-B1 [37,55,56,57]. Besides, two reduced cysteines (Cys251 and Cys384) are present in SR-B1, of which the latter significantly contributes to intrinsic lipid uptake activity of the receptor, and is also necessary for inhibition by BLT-1 [58,59]. Based on our docking result, Cys384 is not included in the optimal binding pose, indicating that inhibition by BLT-1 is more related to the function of SR-B1 rather than blocking the binding site, which is in agreement with conclusion of previous publications [37,58]. Furthermore, the crystal structure of human LIMP-2 (a structural homologue of SR-B1), as reported by Neculai et al. [23], confirmed that interconnected cavities form a tunnel, through which ligands such as cholesterols are delivered to membrane. They also reported that residues of Glu93, Arg95, Lys97, Lys115, Asp252, Asp254, Lys381 and Glu413, in LIMP-2 monomer, are tightly clustered, forming a network that contributes to lining of the tunnel cavity. In the same study, it was further reported that Val214, Val268, Val277, Met337, Phe339, Ile376, Ala379 and Lys381 are present in the cholesterol-binding site of LIMP-2, which facilitates the same interface as SR-B1 for ligand binding [60]. Interestingly, these sites are quite close to the residues that interact with vitamin K and PEG through our docking simulations using SR-B1. In conclusion, based on the molecular docking data, the SR-B1 traps not only cholesterol and lipid molecules such as vitamin K but most likely also PEGylated lipid-based mixed micelles. The small size of the micelles (size 7 to 11 nm, Table 1) is a special advantage. The width of microvillar channel of SR-B1 is estimated to be 8–28 nm [61,62], allowing for the transfer of the vitamin-K-loaded micelles from the extracellular environment to the cytosol. 

Together, our data demonstrate that PEGylation not only increases the colloidal stability of vitamin-K-loaded micelles at low pH but also assisted in SR-B1 mediated internalization. Many studies regarding the intracellular transport pathway demonstrate that lipophilic molecules such as cholesterol and vitamin K are transported as single molecules through the endoplasmic reticulum (ER) and golgi apparatus and are then packed into chylomicrons [63]. After exocytosis from enterocytes, these chylomicrons are mainly transported to lacteal and subsequently enter the systemic circulation via lymphatic vessels and the thoracic duct [64,65]. In our previous study, vitamin K was indeed detected in chylomicrons after internalization of vitamin-K-loaded mixed micelles with and without PEGylation by Caco-2 cell monolayers [9]. The integrity of PEGylated micelles during the internalization and transport process needs further investigation, and therefore the exact route of vitamin K transport remains to be resolved. Specifically, it remains to be determined through which kind of intracellular pathway the micelles and/or vitamin K are packed into chylomicrons and whether this process is affected when vitamin K is taken up in/from PEGylated micelles.

## 4. Conclusions

In the current study, data are presented that convincingly demonstrate the involvement of SR-B1 in the binding, internalization, and transport of vitamin-K-loaded micelles with and without PEG decoration. SPR analysis provided evidence that SR-B1 is able to interact with PEGylated micelles, which is supported by molecular docking simulations, providing a new understanding beyond the commonly known steric effect and hydrophilicity of PEGs in resisting protein adsorption. An optimal PEG density provides the colloidal stability of the micelles while allowing for the interaction with the receptor, which in turn depends on the conformation of the PEG chains in the shell of the micelles. We propose that PEGylated mixed micelles with a size < 20 nm thus allow transport across intestine facilitated by SR-B1. It will be interesting to study if this approach is applicable to the oral uptake of other poorly soluble drugs.

## Figures and Tables

**Figure 1 pharmaceutics-13-02022-f001:**
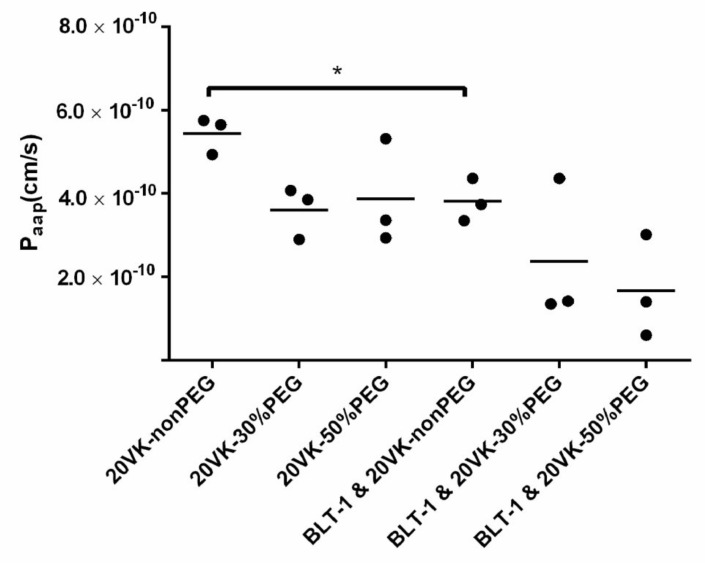
Permeability coefficients of vitamin K (350 μM) loaded in mixed micelles (non-PEG, 30%-PEG and 50%-PEG with the lipid concentration of 1.58 mM) in differentiated Caco-2 cells with and without pre-incubation with 10 μM BLT-1, shown as mean of triplicates for each group (two-tail unpaired *t*-test, * *p* ˂ 0.05).

**Figure 2 pharmaceutics-13-02022-f002:**
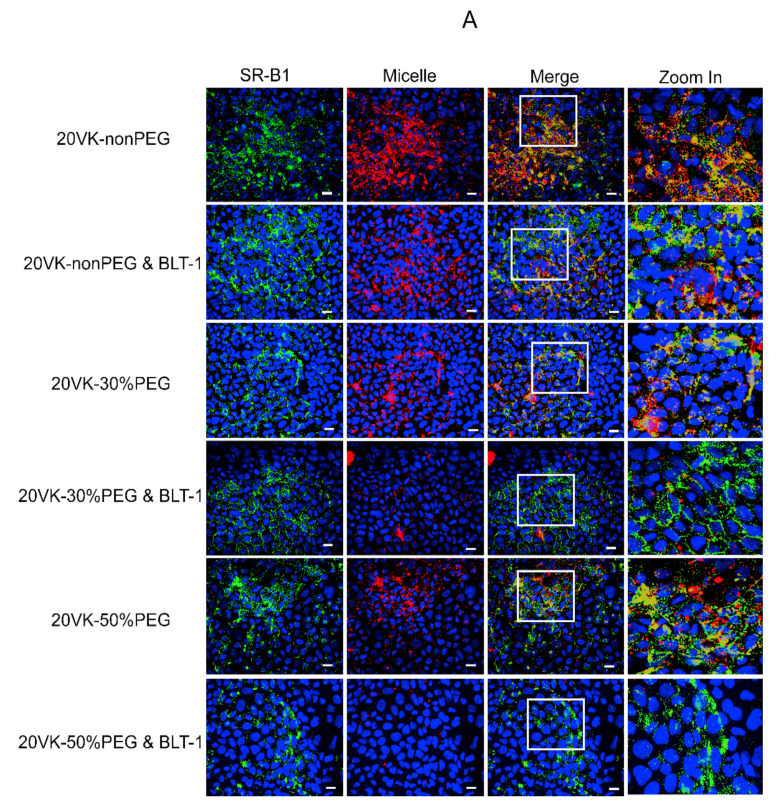
(**A**) Confocal fluorescence microscopy pictures of differentiated Caco-2 cells incubated for 2 h at 37 °C with rhodamine-labeled mixed micelles (red) and immunofluorescence staining (green) of SR-B1, with and without pre-treatment with 10 μM BLT-1. Corresponding fluorescence intensity of rhodamine (**B**) and Pearson’s coefficients of red and green fluorescence (**C**) to reflect the colocalization degree of SR-B1 with mixed micelles; data shown as mean ± SD (*n* = 3, two-tail unpaired *t*-test, * *p* ˂ 0.05, ns is not significant).

**Figure 3 pharmaceutics-13-02022-f003:**
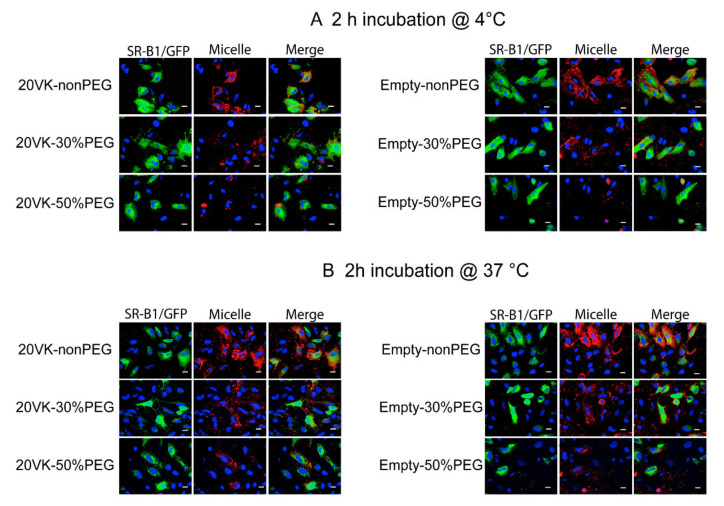
Representative confocal microscopic images of Hela cells overexpressing SR-B1/GFP (green) upon incubation at 4 (**A**) and 37 °C (**B**) with mixed micelles (from top to bottom: non-PEG, 30%-PEG, and 50%-PEG, with the lipid concentration of 1.58 mM) without (empty) and with vitamin K (350 μM). Micelles were labeled with rhodamine (red); scale bar is 20 µm.

**Figure 4 pharmaceutics-13-02022-f004:**
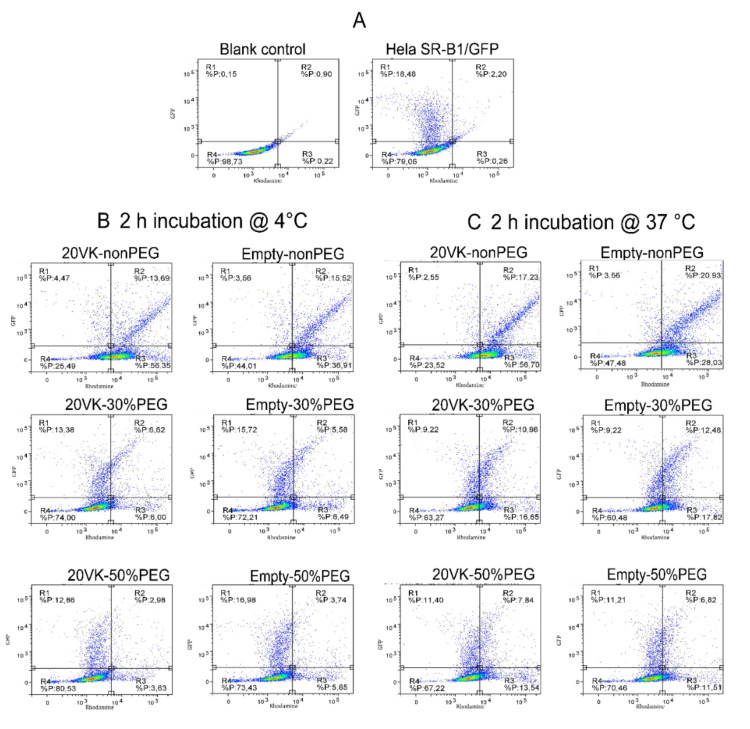
FACS analysis of Hela cells showing SR-B1/GFP fluorescence (*Y*-axis) and rhodamine fluorescence (*X*-axis): (**A**) non-transfected (left) and transfected (right) Hela cells not exposed to mixed micelles; (**B**,**C**) incubation of transfected cells, with from top to bottom: non-PEGylated, 30%-PEGylated and 50%-PEGylated mixed micelles (1.58 mM lipid) labeled with rhodamine, without (empty) or with vitamin K loading (350 μM) at 4 °C (**B**), and at 37 °C (**C**). R1 + R2: Hela cells with SR-B1/GFP signal, R3 + R4: Hela cells without SR-B1/GFP signal; (**D**,**E**) quantification of corresponding fluorescence of rhodamine for transfected HeLa cells incubated with micellar formulations, data shown as mean ± SD, *n* = 3. The quantified fluorescence intensity was normalized for the intensity from 20VK-50PEG micelles in cells without SR-B1/GFP. (Two-way ANOVA, **** *p* ˂ 0.0001, *** *p* ˂ 0.001, ** *p* ˂ 0.01, * *p* ˂ 0.05, ns is not significant).

**Figure 5 pharmaceutics-13-02022-f005:**
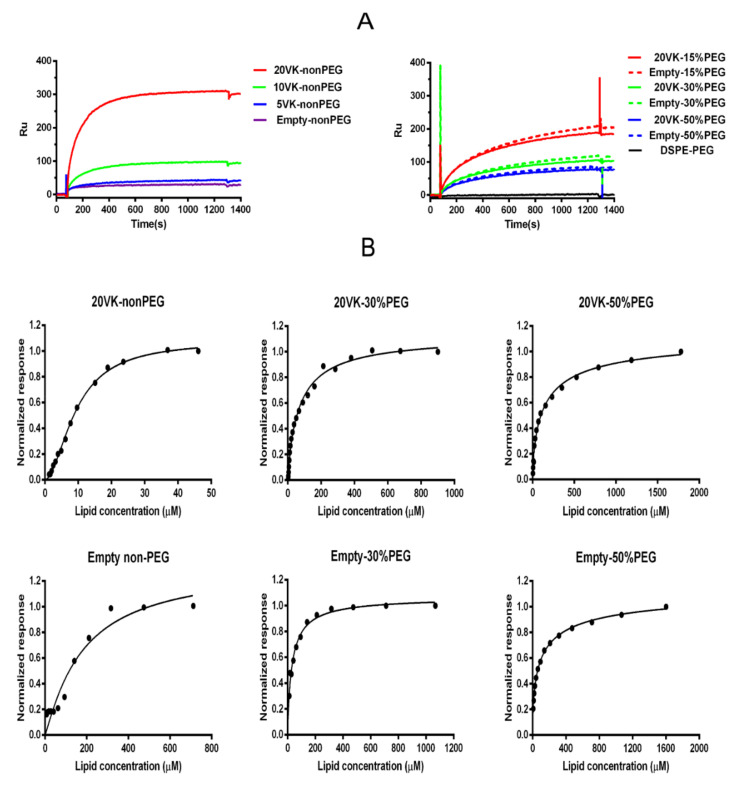
SPR analysis of the binding of mixed micelles to SR-B1, with varying vitamin K content (mg of VK) and PEG density (% of EPC replaced by DSPE-PEG). (**A**) Binding response of mixed micelles (non-PEG, 15%, 30% and 50%-PEG, with and without vitamin K, with empty DSPE-PEG micelle as control) at a lipid concentration of 36 μM. (**B**) Dose-response curves of serial dilutions of mixed micelles, expressed as total lipid concentration, normalized by the maximal response signal at saturation.

**Figure 6 pharmaceutics-13-02022-f006:**
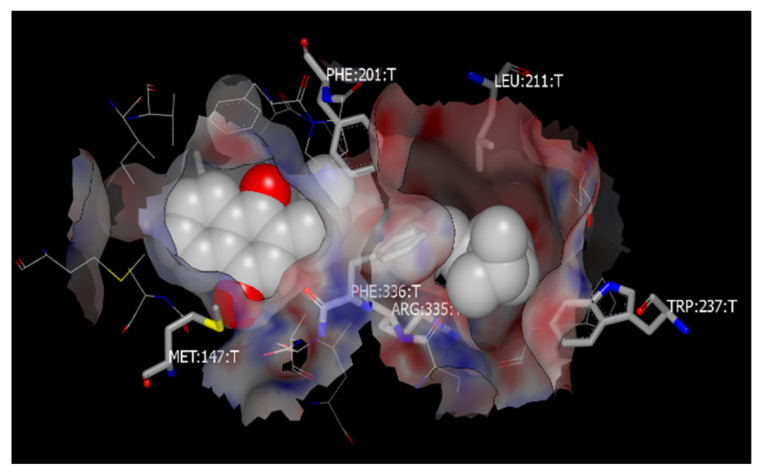
A selected low energy docking pose of vitamin K (White) with SR-B1, where numerous lipophilic residues are present to interact with vitamin K, similarly to the native ligand cholesterol.

**Table 1 pharmaceutics-13-02022-t001:** Characteristics of mixed micelles with varying vitamin K loading and PEG content.

Formulations *	EPC/DSPE-PEG (Total 25 mM)	Glyco- Cholic Acid (mM)	Vitamin K (mM)	Encapsulation Efficiency (%)	Z-Average Diameter (nm)	Zeta Potential (mV)
20VK-nonPEG	100/0	30	5.55	95 ± 2	7.5 ± 0.2	−20.6 ± 2.2
10VK-nonPEG	100/0	30	2.78	91 ± 3	7.3 ± 0.1	−18.9 ± 0.6
5VK-nonPEG	100/0	30	1.39	96 ± 2	7.3 ± 0.1	−17.7 ± 0.5
Empty-non-PEG	100/0	30	0	-	7.4 ± 0.1	−20.4 ± 1.3
20VK-15%PEG	85/15	30	5.55	91 ± 2	9.1 ± 0.1	−14.7 ± 1.1
Empty-15%PEG	85/15	30	0	-	8.4 ± 0.1	−12.9 ± 0.7
20VK-30%PEG	70/30	30	5.55	93 ± 2	9.7 ± 0.6	−9.2 ± 0.7
Empty-30%PEG	70/30	30	0	-	8.7 ± 0.1	−8.5 ± 0.8
20VK-50%PEG	50/50	30	5.55	91 ± 4	10.7 ± 0.4	−5.1 ± 0.2
Empty-50%PEG	50/50	30	0	-	10.5 ± 0.1	−7.3 ± 0.6
DSPE-PEG	0/100	0	0	-	14.1 ± 0.1	−5.9 ± 0.7

Average ± SD of 3 independently prepared batches; * xxVK indicates the weight (mg) of vitamin K present. In the formulations, “Empty” represents micelles without vitamin K.

**Table 2 pharmaceutics-13-02022-t002:** Compositions in weight percentages of the micelles used for SPR analysis, and the corresponding K_D_ and B_max_ (mean ± SD, *n* = 3).

Formulations	EPC (%)	DSPE-PEG (%)	Glycocholic Acid (%)	Vitamin K (%)	K_D_ (μM)	B_max_ (Ru)
20VK-nonPEG	55.2	0	37.9	6.6	9.2 ± 0.1	408 ± 25
10VK-nonPEG	57.1	0	39.3	3.3	-	-
5VK-nonPEG	58.2	0	40	1.8	-	-
Empty-nonPEG	59.3	0	40.7	0	159 ± 19	168 ± 13
20VK-15%PEG	38.6	24.5	31.2	5.4	-	-
Empty-15%PEG	40.9	26	33.1	0	-	-
20VK-30%PEG	27	41.7	26.5	4.7	59 ± 9	408 ± 35
Empty-30%PEG	28.4	43.8	27.8	0	37 ± 7	188 ± 27
20VK-50%PEG	16.1	57.8	22.1	3.6	133 ± 34	319 ± 14
Empty-50%PEG	16.7	60.3	23	0	128 ± 25	226 ± 19
DSPE-PEG	0	100	0	0	-	-

## Data Availability

The data presented in this study are available on request from the corresponding author.

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
