# Peer review of "Internalization and Transport of PEGylated Lipid-Based Mixed Micelles across Caco-2 Cells Mediated by Scavenger Receptor B1"

_pharmaceutics, 2021, doi:10.3390/pharmaceutics13122022_

Round 1

Reviewer 1 Report

This manuscript has clearly demonstrated the PEGylation influence on cell uptakes via Scavenger Receptor B1. Here are few suggestions:

1, The non-PEGylation mixed micelles blended with various concentrations of PEG-DSPE were suggested to treat with the Caco-2 cells to identify the PEG role in internalization.

2.Some minor modifications were required check. For example,  "   2.9 Molecular docking to study the binding sites of SR-B1 with vitamin K1 and PEG. " K1 should be K1. 

Author Response

1, The non-PEGylation mixed micelles blended with various concentrations of PEG-DSPE were suggested to treat with the Caco-2 cells to identify the PEG role in internalization.

Response: That’s indeed what we did in this manuscript. We prepared mixed micelles with different PEG density (by replacing EPC with DSPE-PEG, can also say various concentrations of DSPE-PEG) from 0% to 100% as shown in Table 1. We treated Caco-2 cells by non-PEG, 30% PEGylated, 50% PEGylated mixed micelles for transport experiments (Figure 1) and colocalization (Figure 2). Together with other results in this manuscript, we think it’s clear to identify the role of PEG in internalization.

2.Some minor modifications were required check. For example,  "   2.9 Molecular docking to study the binding sites of SR-B1 with vitamin K1 and PEG. " K1 should be K1. 

Response: Reviewer is right for the subscript 1 to identify vitamin K, however we corrected this by vitamin K without subscript, to be consistent with whole manuscript and our previous publications.

Besides, we made other minor modifications including the source information of the cell line used in this manuscript that has been added to section 2.3. Also, an error of missing equation (6) on page 17 has been corrected.

Reviewer 2 Report

On request of Pharmaceutics, I reviewed the paper  entitled “Internalization and transport of PEGylated lipid-based mixed micelles across Caco-2 cells mediated by Scavenger Receptor B1” by Su et al. The work is an elongation of two previous ones, the first dealt with the preparation of mixed micelles loaded with vitamin K and the second one with their uptake from Caco-2 cells. Here, the Authors investigated the mechanism of internalization of the micelles by studying the role of Scavenger Receptor B1. The work is well structured  and the results all clearly exposed in the text. However, a few adjustments are required prior publication in Pharmaceutics.

Specific comments

  1. The abstract needs a significant revision as it is very confusing and disorganized. First of all it is extremely long (388 words) instead of 200. Besides, it describes a previous paper creating misunderstanding while on the contrary detailed data acquired in the current study are poorly provided. I suggest the Authors to underline, the significance or not significance of the data obtained from the different formulations, otherwise the conclusions are not supported.
  2. Please transfer the dot after the citation number.
  3. In Table 1 to better evidence the pre-formulation study results, it would be of paramount importance to add encapsulation efficiency % and drug loading.
  4. The Authors stated “Dispersions were 20 times diluted with 10 mM HEPES buffer (pH 7.4) and the zeta-potential of the different micelles was determined”. Please provide an explanation of the Zeta potential data. Why did you perform the zeta potential only in a saline buffer? Why are the measurements in Hepes buffer so negative? Please provide in SI an original representative size and zeta potential distribution, and preferably add the effective net charge in water.
  5. The Authors stated “increasing PEG content the zeta potential at pH 7.4 became less negative from -20.6 ± 2.2 mV to -5.1 ± 0.2 mV, which can be ascribed to shielding of the negatively carboxylate ani-ons of glycolic acid by the PEG chains” please provide some references supporting this speculation. Indeed I think that the DSPE-PEG itself is responsible for less negative Zeta potential being its value -6 mV. How did the PEG chain exert repulsion or electrostatic shielding?

Author Response

  1. The abstract needs a significant revision as it is very confusing and disorganized. First of all it is extremely long (388 words) instead of 200. Besides, it describes a previous paper creating misunderstanding while on the contrary detailed data acquired in the current study are poorly provided. I suggest the Authors to underline, the significance or not significance of the data obtained from the different formulations, otherwise the conclusions are not supported.

Response: The abstract has been revised to 211 words to focus more on the interaction of PEGylated mixed micelles with SR-B1, which is the core concept of the whole manuscript.

  1. Please transfer the dot after the citation number.

Response: It’s corrected in the revised manuscript.

  1. In Table 1 to better evidence the pre-formulation study results, it would be of paramount importance to add encapsulation efficiency % and drug loading.

Response: Encapsulation efficiency (EE%) and vitamin K content (drug loading) are added in Table 1 and Table 2, the corresponding description and equations were added in materials and methods section (page 3, end of section 2.2) and results and discussion (page 8, section 3.1).

Consequently, the numbers of equations in whole manuscript were re-arranged.

  1. The Authors stated “Dispersions were 20 times diluted with 10 mM HEPES buffer (pH 7.4) and the zeta-potential of the different micelles was determined”. Please provide an explanation of the Zeta potential data. Why did you perform the zeta potential only in a saline buffer? Why are the measurements in Hepes buffer so negative? Please provide in SI an original representative size and zeta potential distribution, and preferably add the effective net charge in water.

Response: Size and zeta potential distributions are provided in supporting information Figure S1 of the revised manuscript. Correspondingly, numbers of figures in supporting information and manuscript were re-arranged.

The negative charge is mainly caused by deprotonated carboxylic acid groups of glycocholic acid (with pKa of 4.4) in the buffer with pH 7.4, as was mentioned in section 3.1 of the manuscript.              

About the explanation of the Zeta potential, please refer to reference 28 (Bhattacharjee, S., "DLS and zeta potential - What they are and what they are not?" J Control Release 2016, 238, 337-351). Zeta potentials should always be measured in a buffer with low ionic strenght, because pH of molecules or colloidal particles in pure water is not defined. In fact, zeta potentials are depending on both pH as well as ionic strenght. Therefore both should be well defined. A zwitterionic solution such as HEPES buffer with low ionic strength is routinely used for Zeta potential measurement. There is no physical sense of the existence of zeta potential of pure water.

  1. The Authors stated “increasing PEG content the zeta potential at pH 7.4 became less negative from -20.6 ± 2.2 mV to -5.1 ± 0.2 mV, which can be ascribed to shielding of the negatively carboxylate ani-ons of glycolic acid by the PEG chains” please provide some references supporting this speculation. Indeed I think that the DSPE-PEG itself is responsible for less negative Zeta potential being its value -6 mV. How did the PEG chain exert repulsion or electrostatic shielding?

Response: The low zeta potential of mixed micelles without PEGylation composed of bile salt and EPC is caused by deprotonation of the COOH groups of glycocholic acid, as explained above and in the manuscript. DSPE-PEG is slightly electronegativity, but larger PEG and higher density can effectively shield the negative charge, as was explained in section 3.1 of the manuscript. Please refer to reference 30 and 31 in the manuscript (Stamp, D.H. Three hypotheses linking bile to carcinogenesis in the gastrointestinal tract: certain bile salts have properties that may be used to complement chemotherapy. Medical Hypotheses 2002, 59, 398-405.Tatiana S. Levchenko, R.R., Anatoly N. Lukyanov, Kathleen R. Whiteman, Vladimir P. Torchilin. Liposome clearance in mice: the effect of a separate and combined presence of surface charge and polymer coating. International Journal of Pharmaceutics 2002, 95–102.)

Regarding the PEG chain repulsion or electrostatic shielding, please read reference 43 and 44 (Rabanel, J.M.; Hildgen, P.; Banquy, X. Assessment of PEG on polymeric particles surface, a key step in drug carrier translation. J Control Release 2014, 185, 71-87.Zhou, H.; Fan, Z.; Li, P.Y.; Deng, J.; Arhontoulis, D.C.; Li, C.Y.; Bowne, W.B.; Cheng, H. Dense and dynamic polyethylene glycol shells cloak nanoparticles from uptake by liver endothelial cells for long blood circulation. ACS Nano. 2018, 12, 10130-10141.), PEG coatings are well known to prevent aggregation and to stabilize particles and colloidal suspensions in physiological salt concentration media by steric and hydration repulsions.

Reviewer 3 Report

The authors fabricated mixed micelles composed of EPC, DSPE-PEG, glycocholic acid loaded with vitamin K. They tried to investigate the transport mechanism of the mixed micelles with different PEG densities using a variety of sensible techniques. In my opinion, the manuscript deserves publication after minor revisions.

Minor revisions:

  • You better insert scale bars in all confocal fluorescent microscopy images of Figure 2A, Figure 3A, B, and Figure S3.
  • What is the purpose of incubating test cells at 4℃ in the entire binding cellular uptake experiments?
  • I believe that the binding and cellular uptake of the mixed micelles is time-dependent. Why do you only stick to 2 h? It would be very informative if we see the binding and uptake trends at 30 min and 1 h also.

Author Response

  • You better insert scale bars in all confocal fluorescent microscopy images of Figure 2A, Figure 3A, B, and Figure S3.

Response: Scale bars are inserted in all confocal fluorescent microscopy images of Figure 2A, Figure 3A, B, and Figure S3 (now is Figure S4)

  • What is the purpose of incubating test cells at 4℃ in the entire binding cellular uptake experiments?

Response: The purpose of incubating cells at 4℃ is to inhibit active internalization (which requires energy), then to study the binding affinity of the formulations to the receptor on the surface of the cells, as was explained in section 3.4 (second paragraph) of the manuscript.

  • I believe that the binding and cellular uptake of the mixed micelles is time-dependent. Why do you only stick to 2 h? It would be very informative if we see the binding and uptake trends at 30 min and 1 h.

Response: The binding and uptake of mixed micelles is indeed time-dependent, which was shown in our previously published paper (Sun et al., Influence of PEGylation of vitamin-K-loaded mixed micelles on the uptake by and transport through Caco‑2 Cells. Mol. Pharmaceutics 2018, 15, 3786−3795). The purpose of the present work was to compare binding and uptake between different formulations. Considering to obtain sufficient signal for lower affinity formulations and condition of cells incubated at 4℃, here 2h incubation is sufficient to study and compare binding and uptake.